# POSTERIOR-GRPO: REWARDING REASONING PROCESSES IN CODE GENERATION

## ABSTRACT

Reinforcement learning (RL) has significantly advanced code generation for large language models (LLMs). However, current paradigms rely on outcome-based rewards from test cases, neglecting the quality of the intermediate reasoning process. While supervising the reasoning process directly is a promising direction, it is highly susceptible to reward hacking, where the policy model learns to exploit the reasoning reward signal without improving final outcomes. To address this, we introduce a unified framework that can effectively incorporate the quality of the reasoning process during RL. First, to enable reasoning evaluation, we develop LCB-RB, a benchmark comprising preference pairs of superior and inferior reasoning processes. Second, to accurately score reasoning quality, we introduce an Optimized-Degraded based (OD-based) method for reward model training. This method generates high-quality preference pairs by systematically optimizing and degrading initial reasoning paths along curated dimensions of reasoning quality, such as factual accuracy, logical rigor, and coherence. A 7B parameter reward model with this method achieves state-of-the-art (SOTA) performance on LCB-RB and generalizes well to other benchmarks. Finally, we introduce Posterior-GRPO (P-GRPO), a novel RL method that conditions process-based rewards on task success. By selectively applying rewards to the reasoning processes of only successful outcomes, P-GRPO effectively mitigates reward hacking and aligns the model's internal reasoning with final code correctness. A 7B parameter model with P-GRPO achieves superior performance across diverse code generation tasks, outperforming outcome-only baselines by 4.5%, achieving comparable performance to GPT-4-Turbo. We further demonstrate the generalizability of our approach by extending it to mathematical tasks. Our models, dataset, and code are publicly available.

## 1 INTRODUCTION

Reinforcement learning (RL) has emerged as a transformative post-training paradigm for large language models (LLMs) (Guo et al., 2025; Yang et al., 2025). The recent breakthrough of DeepSeek-R1 (Guo et al., 2025) exemplifies this paradigm shift, achieving remarkable success through techniques like GRPO algorithm (Shao et al., 2024). This advancement has catalyzed extensive research applying RL post-training to enhance code generation in LLMs (Abdin et al., 2025; Zhao et al., 2025; Xu et al., 2025).

Despite these advances, existing approaches exclusively rely on outcome signals from generated code—such as test pass rates—while overlooking the reasoning processes that underlie code quality (Guo et al., 2025; Abdin et al., 2025; Zeng et al., 2025), which may lead to suboptimal reasoning processes, ultimately compromising the accuracy of the final solutions (Zhang et al., 2025). Our preliminary investigation reveals a significant correlation between reasoning process quality and solution correctness ($\chi^2$ test (Pearson, 1900), $p = 9.3 \times 10^{-15} \ll 0.001$; see the correlation analysis in Appendix A.4). Extensive work has also demonstrated that enabling LLMs to simultaneously generate reasoning processes during solution generation can substantially enhance their capabilities (Wei et al., 2022; Lyu et al., 2023). This gap motivates a fundamental research question: *Can we optimize the reasoning processes of policy models to achieve more efficient improvements in their code generation capabilities?*

A typical RL training pipeline for code generation (Guo et al., 2025) follows an intuitive paradigm: the model generates code solutions, receives feedback based on test case outcomes, and updates its policy accordingly. While widely adopted, directly integrating the reasoning process rewards presents three critical challenges:

- Lack of appropriate evaluation benchmarks for assessing reward models' discrimination capabilities on reasoning processes. Existing benchmarks (Liu et al., 2025; Lambert et al., 2024) primarily focus on using solutions rather than reasoning processes, creating misalignment with our objectives.

- Absence of reliable reward models specifically designed for reasoning evaluation. Current state-of-the-art (SOTA) reward models (Liu et al., 2024a; Yuan et al., 2024), such as Skywork-Reward-V2-Llama-3.1-8B (Liu et al., 2024a), are trained on solutions rather than reasoning processes. While correlation exists between reasoning and code quality, the semantic gap between natural language and code structure (Wang et al., 2021) renders direct application of existing models suboptimal.

- Lack of effective RL algorithms that leverage signals from reward models. Research (Guo et al., 2025) has demonstrated that neural reward models may suffer from reward hacking during RL training, particularly in code generation tasks where neural reward model signals are more susceptible to exploitation compared to test case pass rate reward signals.

To address these challenges, we propose a unified training framework that enhances code generation capabilities through reasoning-aware RL. Our core insight is to leverage the intrinsic features of reasoning processes to create reliable training signals that guide policy models toward both correct solutions and high-quality reasoning patterns.

To address the first challenge, we introduce LCB-RB, a benchmark derived from LiveCodeBench (Jain et al., 2025) composed of 187 preference pairs. Each pair consists of a superior and an inferior reasoning process. We then introduce the Optimized-Degraded-based method, which we refer to as the OD-based method, for reward model training to address the second challenge. Specifically, we employ a powerful LLM to generate initial reasoning processes for problems, then transform these into optimized and degraded versions based on three critical dimensions that determine reasoning quality, including factual accuracy, logical rigor and logical coherence. Training on these inherently contrastive pairs enables our reward model to distinguish between high-quality and low-quality reasoning patterns. To address the last challenge, we propose Posterior-GRPO (P-GRPO), a novel algorithm designed to prevent reward hacking while maximizing training signal quality, based on GRPO. Our approach integrates signals from three complementary sources: the thinking reward, outcome reward (i.e., pass rates), and the format reward. Crucially, we employ a posterior reward assignment strategy, in which reasoning rewards are computed only after correct outcomes (i.e., when all test cases pass), ensuring alignment between reasoning quality and functional correctness. An advantage of P-GRPO is its data utilization efficiency, which enables differentiated rewards when all samples are correct, improving the original GRPO's limitation where uniform success yields zero advantage values and no gradient information.

Specifically, for the reward model, we train 3B and 7B parameter models from Qwen2.5-Coder-Base with OD-based method. The 7B model achieves comparable performance to GPT-4-Turbo on LCB-RB while demonstrating strong generalization on the reasoning subset of RewardBench (Lambert et al., 2024). And for RL, P-GRPO effectively enhances the code generation capabilities of Qwen2.5-Coder-7B-Instruct with a relative improvement of 13.9% over the base model (50.4%→57.4%) on LiveCodeBench, HumanEval(+), MBPP(+), and BigCodeBench, surpassing the RL with outcome-only rewards baseline by 4.5% and reaching performance comparable to GPT-4-Turbo. Furthermore, when we extend P-GRPO to mathematical tasks, Qwen2.5-Math-7B achieves a relative improvement of 7.3% over outcome-only reward baselines, demonstrating the generalization capability of our approach.

In summary, our contributions are threefold. First, we introduce LCB-RB, a benchmark designed to evaluate the ability of reward models to discriminate between different levels of reasoning quality. Second, we identify multi-dimensional reasoning features and propose an OD-based method for training thinking reward models. Finally, we present P-GRPO, a novel RL algorithm that leverages process-based rewards to enhance model reasoning capabilities. Our extensive evaluation across 4 code generation benchmarks and 3 mathematical benchmarks demonstrates that P-GRPO effec-

tively improves reasoning performance while exhibiting strong generalization across domains. The models, datasets and code are publicly available[1].

## 2 METHOD

### 2.1 OVERVIEW

While current RL approaches benefit from leveraging outcome rewards, they suffer from a limitation: insufficient consideration of the quality of the LLM's reasoning process. To address this challenge, we first develop a specialized reward model capable of evaluating reasoning processes, then integrate this reward signal with outcome rewards for optimization. Our methodology comprises three distinct stages. We first explain how to design a benchmark tailored for assessing reasoning processes in code generation tasks. Then, we leverage OD-Based method for reward model training. Finally, we introduce how our algorithm P-GRPO integrates thinking reward signals with outcome rewards to optimize the policy model.

### 2.2 LCB-RB BENCHMARK CONSTRUCTION

Although existing benchmarks (Lambert et al., 2024; Liu et al., 2025) include reasoning evaluation subsets that assess reward models' ability to distinguish between correct and incorrect code blocks or mathematical solutions, they are not aligned with our specific objective: evaluating preference-based reasoning processes for code generation.

Following previous work (Lambert et al., 2024; Liu et al., 2025), we leverage a powerful LLM with high-temperature sampling to generate multiple solutions with reasoning processes for each code problem. We construct initial data pairs by selecting reasoning processes from correct and incorrect code solutions. The correctness of each solution is determined by the corresponding test cases. However, high-quality reasoning processes may still lead to erroneous solutions due to the inherent randomness (Wang et al., 2023; 2020) introduced by high-temperature sampling, such as missing crucial import statements. Given that models often struggle to detect their own errors due to self-consistency bias (Bartsch et al., 2023; Liang et al., 2024), we employ the more capable GPT-4o (OpenAI, 2024) as an external validator for further filtering. Specifically, we task the model with two key assessments: (1) identifying whether reasoning processes contain logical flaws, and (2) verifying whether the corresponding implementations faithfully align with the stated reasoning. While GPT-4o could generate solutions, we opt for Qwen2.5-Coder-32B-Instruct for generation due to cost considerations. The task prompt is in Appendix A.7. We retain only instances where code implementation and its reasoning process demonstrate consistency. We then designate reasoning processes from correct code with no logical flaws as chosen examples, while reasoning processes from incorrect code that exhibit identifiable flaws serve as rejected examples. A natural class imbalance emerges when curating LCB-RB. This imbalance stems from an inherent asymmetry in the relationship between code and reasoning in problems that have both correct and incorrect solutions: while a functionally correct code is almost always predicated on a sound reasoning process, an incorrect code does not necessarily indicate a flawed reasoning. For instance, a solution may follow a sound logical flow but fail due to a minor implementation error, such as an import mistake, as stated above. Consequently, the number of chosen examples (sound reasoning, correct code) is larger than the number of rejected examples (flawed reasoning, incorrect code).

To address this, we employ a down-sampling strategy. Specifically, for each rejected example, we randomly sample one corresponding chosen example from the same problem instance to form a preference pair. Problems with only correct or only flawed reasoning processes are excluded, as they cannot be used to form a valid preference pair.

We utilize 880 code problems from LiveCodeBench v5 and employ Qwen2.5-Coder-32B-Instruct with temperature=1.0 to generate 50 solutions with reasoning processes for each problem. The prompt used for generation is the same as that used in RL training in Appendix A.6. Finally, we obtain 187 pairs.

---

[1]https://anonymous.4open.science/r/ReasoningRL-CC6F

## 2.3 OPTIMIZED-DEGRADED BASED METHOD

We introduce the Optimized-Degraded based (OD-based) Method, a novel approach for training reward models that evaluate the quality of reasoning processes. As shown in Figure 1, our method trains a reward model $r^\star$ to assign scalar scores $r^t$ to problem-reasoning pairs $(x, y)$, where each score quantifies the quality of the reasoning path $y$ that leads from a given problem $x$.

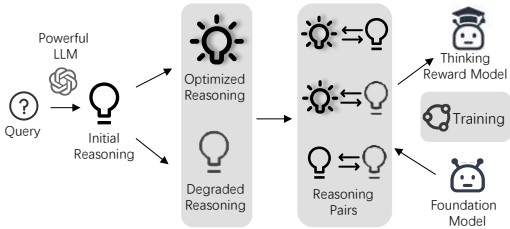

Figure 1: An overview of OD-based method.

The methodology is motivated by the Evol-Instruct paradigm (Xu et al., 2024), which enhances generative models by evolving training data. We hypothesize that this paradigm can be adapted to improve the discriminative capabilities of reward models. Additionally, we introduce a key modification to generate stronger contrastive signals: instead of only optimizing an initial response, we also create a degraded version. Our approach begins by prompting a powerful LLM, Qwen2.5-Coder-32B-Instruct, to generate an initial reasoning process $y$ for a given problem $x$, then generating optimized $(y^+)$ and degraded $(y^-)$ variants by leveraging carefully designed prompts based on three critical dimensions that determine reasoning quality: (1) *Factual Accuracy* assesses whether the reasoning contains factual errors; (2) *Logical Rigor* assesses (a) whether redundant or misleading logical steps exist, and (b) whether missing logical connections result in incomplete reasoning; (3) *Logical Coherence* assesses whether the logical flow maintains clear connections between steps. For instance, to address redundant logical steps, we instruct the model to identify and remove any redundant steps from the initial reasoning. We also prompt the model to select one or more dimensions from our defined set to optimize or degrade the reasoning. The optimization and degradation prompts are in Appendix A.6.

This yields three types of preference pairs: $(x, y^+, y^-)$, $(x, y, y^-)$, and $(x, y^+, y)$. We train on all three types to enable the model to distinguish between high-quality and low-quality reasoning across diverse reasoning patterns. We train a Bradley-Terry (Bradley & Terry, 1952) reward model, which is widely adopted in reward modeling (Ouyang et al., 2022; Bai et al., 2022). Through this process, we obtain a reward model $r^\star$ capable of identifying diverse reasoning patterns and outputting scalar scores proportional to reasoning quality for any given problem-reasoning pair $(x, y)$.

## 2.4 POSTERIOR-GRPO

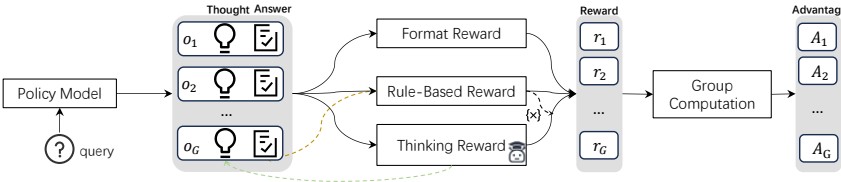

Figure 2: The overview of P-GRPO. It adopts a posterior-based strategy. Specifically, a thinking reward is incorporated into the total reward signal if, and only if, a rule-based reward first confirms the final answer is correct.

Our implementation builds upon GRPO (Shao et al., 2024), an algorithm that enhances policy models by evaluating a group of responses sampled for a given problem. Directly incorporating the thinking reward from a neural reward model makes the policy susceptible to reward hacking (Guo et al., 2025), where it learns to exploit the thinking reward function for high scores without generating better solutions. To address this challenge, we introduce a simple yet effective Posterior-GRPO (P-GRPO) algorithm that leverages reward model outputs more robustly to prevent exploitation, as shown in Figure 2. The reward components are as follows:

(1) Format Reward $R^f$: This binary reward ensures structural compliance by verifying whether the model's output adheres to the prescribed format—specifically, encapsulating reasoning within "<think>...</think>" and answers within "<answer>...</answer>". We assign $R^f = 1$ if the

format is correct, and $R^f = 0$ otherwise. It is used to ensure reliable extraction, which has been demonstrated to be effective in prior works (Xie et al., 2025; Guo et al., 2025).

(2) Rule-Based Reward $R^o$: This reward is derived from test case-based verification of the extracted answer's correctness; this binary reward employs strict evaluation criteria. We set $R^o = 1$ only when the extracted result passes all test cases; otherwise, $R^o = 0$.

(3) Thinking reward $R^t$: This reward is generated by a thinking reward model that evaluates the quality of the extracted reasoning process. It is continuous and ranges from 0 to 1. To mitigate reward hacking, we implement a posterior-based adjustment mechanism. Specifically, when the outcome reward $R^o = 1$, the thinking reward $R^t$ is preserved as the final reasoning score. However, when $R^o \neq 1$, we set $R^t = 0$. This gated design ensures that the model is only incentivized to explore superior reasoning paths for solutions that are functionally correct, while simultaneously preventing it from exploiting the thinking reward on incorrect outcomes. This mitigates reward hacking and ensures the model's internal optimization process is aligned with final solution correctness. The final reward computation integrates these components as follows:

$$R_i = R_i^f + R_i^o + R_i^o \cdot R_i^t \tag{1}$$

Furthermore, in standard GRPO (Shao et al., 2024), when all samples in a batch are correct, this uniform success results in zero advantage values and erases gradient information. P-GRPO overcomes this by augmenting the rule-based reward with thinking reward. Even when all solutions in a batch are functionally correct, their underlying reasoning paths can vary in quality, thus yielding different total rewards. This introduces meaningful differentiation into the reward signal, creating non-zero advantage values. As a result, the model receives a clear gradient signal to not only find a correct solution but to actively prefer solutions derived from high-quality reasoning.

## 3 EXPERIMENTAL SETUP

**Reward Model Setup** We evaluate our reward model on LCB-RB and the code and math subsets of RewardBench (Lambert et al., 2024), using accuracy as the evaluation metric (Lambert et al., 2024; Liu et al., 2025). Our models are initialized from Qwen2.5-Coder-7B-Base and Qwen2.5-Coder-3B-Base, and trained on preference pairs from the DeepCoder-Preview-Dataset (Luo et al., 2025), a corpus of 24k coding problems. We compare our approach against several baselines: the original base models, state-of-the-art (SOTA) reward models (including Starling-RM-34B (Zhu et al., 2023), EURUS-RM-7B (Yuan et al., 2024), Skywork-Reward-Llama-3.1-8B (Liu et al., 2024a), GPT-4-Turbo-2024-04-09 (Achiam et al., 2023), and GPT-3.5-Turbo-0125 (OpenAI, 2022)), and a Score-Based reward model. Further details are provided in appendix A.1.

**RL Setup** We conduct evaluations on HumanEval(+) (Liu et al., 2024b; Chen et al., 2021), MBPP(+) (Austin et al., 2021; Liu et al., 2024b), BigCodeBench (Zhuo et al., 2024), and Live-CodeBench v5(Jain et al., 2025). We use greedy decoding and employ Pass@1 metric. We select Qwen2.5-Coder-7B-Instruct as policy model, using DeepCoder-Preview-Dataset (Luo et al., 2025) for training. Following recent work (Yu et al., 2025; He et al., 2025; Wang et al., 2025a) on stabilizing RL, we remove the KL divergence term and adopt the clip-higher and token-level loss strategy. The prompt used for training is in Appenidx A.3. Baselines for comparison include the original model, SOTA code models (including Llama3-Instruct-70B (AI@Meta, 2024), Deepseek-Coder-V2-Lite-Instruct (Zhu et al., 2024), Qwen2.5-Coder-Instruct 14B (Hui et al., 2024), GPT-4-Turbo-2024-04-09 (Achiam et al., 2023), and GPT-3.5-Turbo-0125 (OpenAI, 2022)), the model fine-tuned with SFT on the same RL data, and the model only with outcome and format rewards. Further details provided in Appendix A.2.

## 4 RESULTS

In this section, we aim to answer the following research questions:

- **RQ1**: How effective is our approach, P-GRPO, in improving code generation across different benchmarks?
- **RQ2**: How effectively does the reward model with the OD-based method discriminate between high-quality and low-quality reasoning processes on LCB-RB? Furthermore, does this discriminative capability generalize to other reasoning benchmarks?

| Model | Size | Humaneval | | MBPP | | LiveCodeBench | | | BigCodeBench | | Average |
| | | HE | HE+ | MBPP | MBPP+ | Easy | Medium | Hard | Full | Hard | - |
|---|---|---|---|---|---|---|---|---|---|---|---|
| GPT-4-Turbo | 🔒 | 90.2 | 86.0 | 85.7 | 73.3 | 68.5 | 24.2 | 4.6 | 58.2 | 35.1 | 58.4 |
| GPT-3.5-Turbo | 🔒 | 72.6 | 67.7 | 84.1 | 71.2 | 46.3 | 9.4 | 5.6 | 50.6 | 21.6 | 47.7 |
| Qwen2.5-Coder-Instruct | 14B | 89.6 | 87.2 | 86.2 | 72.8 | 61.0 | 11.3 | 2.8 | 48.4 | 22.2 | 53.5 |
| DS-Coder-V2-Lite-Instruct | 2.4/16B | 81.1 | 75.6 | 82.8 | 70.4 | 43.9 | 5.7 | 5.6 | 36.8 | 16.2 | 46.5 |
| Llama3-Instruct | 70B | 77.4 | 72.0 | 82.3 | 69.0 | 43.9 | 7.5 | 5.6 | 54.5 | 27 | 48.8 |
| Qwen2.5-Coder-Instruct | 7B | **88.4** | **84.1** | 83.5 | 71.7 | 56.1 | 3.8 | 6.9 | 41.0 | 18.2 | 50.4 |
| +SFT | 7B | 66.2 | 57.3 | 73.3 | 63.5 | 34.1 | 3.8 | 0.0 | 39.9 | 13.5 | 39.1 |
| +GRPO (Code) | 7B | 85.9 | 81.1 | 86.7 | 75.1 | 58.5 | 15.1 | **9.7** | 52.0 | 29.7 | 54.9 |
| +P-GRPO (Code+RM) | 7B | 86.6 | 81.1 | **87.0** | **76.2** | 68.3 | 20.8 | 9.7 | 54.0 | **33.8** | **57.4** |

Table 1: Performance comparison of Qwen2.5-Coder with P-GRPO against other baselines on code generation benchmarks.

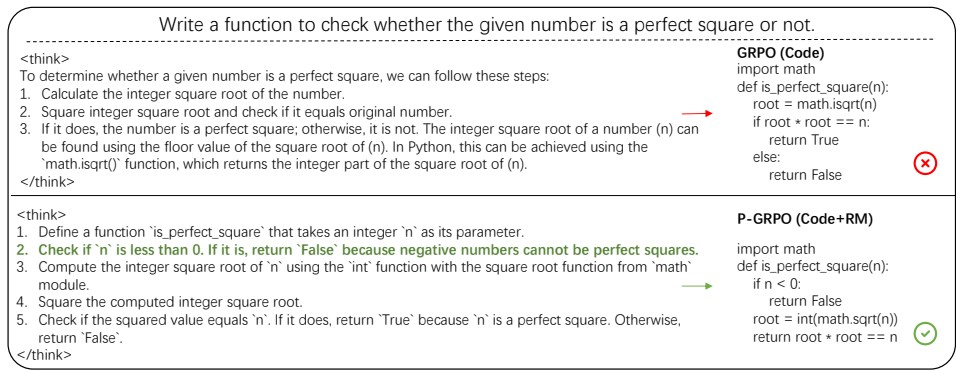

Figure 3: Example of reasoning processes generated by the base model with P-GRPO and with GRPO.

- **RQ3**: Can the P-GRPO training paradigm generalize to mathematical tasks which also rely on high-quality reasoning capabilities?

## 4.1 RQ1: EFFECTIVENESS OF P-GRPO IN CODE GENERATION

As presented in Table 1, results demonstrate that P-GRPO effectively enhances the performance of the model, achieving SOTA performance in code generation. P-GRPO achieves a relative improvement of 13.9% on average over the base model across all benchmarks. Additionally, it surpasses the baseline with outcome-only rewards by 4.5% on average, with this gain being pronounced on LiveCodeBench (18.1% relative improvement), showing comparable performance to GPT-4-Turbo.

Compared with the model without thinking reward, the model with P-GRPO demonstrates consistent superiority. As shown in Figure 4a, the performance of the base model with P-GRPO consistently outperforms the baseline. To further elucidate the mechanisms underlying P-GRPO's performance gains, we manually inspect models' outputs. Our analysis reveals that P-GRPO's primary advantage lies in its ability to generate more comprehensive and logically sound reasoning processes, which help the model produce more accurate code. For example, as shown in Figure 3, when solving a perfect square problem, the model without thinking reward fails to consider negative numbers as edge cases in its reasoning process, causing pass only the basic test cases while failing the test cases in MBPP+. In contrast, the model with P-GRPO demonstrates sound reasoning by considering negative inputs at the outset of its reasoning process. As shown in Figure 3, the results show that P-GRPO achieves a thinking reward score of 0.21, outperforming the baseline without thinking reward, which obtains a score of 0.02. This highlights how a well-formed reasoning path contributes to a more accurate final output. More examples are in supplementary materials. We also find that RL yields superior performance compared to SFT. This suggests that further SFT may disrupt a model's established capabilities, whereas RL allows for more targeted policy refinement through exploration (Chu et al., 2025).

| Model | Size | LCB-RB - | RewardBench Code | RewardBench Math | Avg - |
|---|---|---|---|---|---|
| GPT-4-Turbo | 🔒 | 58.28 | 98.07 | 67.34 | 74.56 |
| GPT-3.5-Turbo | 🔒 | 50.53 | 77.64 | 40.60 | 56.26 |
| Starling-RM | 34B | 52.40 | 88.82 | 85.90 | 75.71 |
| EURUS-RM | 7B | 56.68 | 92.78 | 79.86 | 76.44 |
| Skywork -Reward -Llama-3.1 | 8B | 57.75 | - | - | - |
| Qwen2.5-Coder | 3B | 49.19 | 52.84 | 59.95 | 53.99 |
| +Score-Based | 3B | 52.10 | 49.39 | 47.20 | 49.56 |
| +OD-Based | 3B | 52.40 | 63.61 | 93.51 | 69.84 |
| Qwen2.5-Coder | 7B | 51.33 | 43.90 | 65.77 | 53.67 |
| +Score-Based | 7B | 47.59 | 80.18 | 71.81 | 66.53 |
| +OD-Based | 7B | **58.28** | **88.61** | **99.77** | **82.22** |

Table 2: Performance comparison of reward model trained with OD-based method against other baselines.

| Model | Size | MATH 500 | Minerva Math | AIME24 | Avg |
|---|---|---|---|---|---|
| GPT-4o | 🔒 | 76.4 | 36.8 | 9.3 | 40.8 |
| Llama-3.1-Inst | 70B | 64.6 | 35.3 | 16.7 | 38.9 |
| Llama-3.1-Inst | 405B | 73.8 | 54.0 | 20.0 | 49.3 |
| Eurus-2-PRIME | 7B | 79.2 | 38.6 | 26.7 | 48.2 |
| Qwen2.5-Math-Inst | 7B | 79.8 | 37.1 | 13.3 | 43.4 |
| Qwen2.5-Math | 7B | 46.9 | 15.5 | 11.2 | 24.5 |
| +GRPO | 7B | **83.0** | 34.2 | 26.7 | 48.0 |
| +P-GRPO | 7B | **83.0** | **38.2** | **33.3** | **51.5** |

Table 3: Performance comparison of Qwen2.5-Math with P-GRPO against other baselines on various math benchmarks.

## 4.2 RQ2: Reward Model Effectiveness

The results are in Table 2. Due to potential data contamination, we exclude the results of Skywork-Reward-Llama-3.1 from RewardBench[2]. The reward model trained with OD-based method effectively enhances the base model's ability to identify high-quality reasoning processes on LCB-RB, surpassing all other baselines. For instance, our 7B parameter model achieves a relative improvement of 10.2% over GPT-4-Turbo and 23.5% over the score-based baseline.

Compared with the score-based method, the OD-based method demonstrates substantial improvements. Specifically, our 3B and 7B models achieve relative improvements of 40.9% and 23.5%, respectively, on selected benchmarks. This finding indicates that training LLMs to distinguish between optimized and degraded versions of reasoning processes is more effective than learning from direct numerical scores. This is likely because LLMs are not inherently sensitive to fine-grained numerical values (Feng et al., 2024; Ahn et al., 2024), making it difficult to express the nuanced differences between reasoning processes via a scalar score. Furthermore, the base model trained with OD-based method demonstrates SOTA performance on reasoning subsets of RewardBench, outperforming the best baseline by 7.8% relatively in accuracy. This result underscores that the quality of the intermediate reasoning process influences the quality of the final solution (Wei et al., 2022).

## 4.3 RQ3: Generalization to Mathematical Tasks

To further assess the generalization of our approach, we extend our framework to the mathematical domain where performance hinges critically on high-quality reasoning.

**Experimental Setup** We employ the same reward model used in RQ1. For the policy model, we utilize Qwen2.5-Math-7B (Yang et al., 2024), a base model for mathematical tasks that has been used for RL in previous work (Wang et al., 2025b; Zuo et al., 2025). We choose DAPO-Math-17k (Yu et al., 2025) as the training dataset, which consists of 17K mathematical data. We maintain the same experimental setup as in RQ1, except the training steps are reduced to 900, accounting for the smaller dataset size. Model performance is evaluated on MATH500 (Hendrycks et al., 2021), Minerva Math (Lewkowycz et al., 2022) and AIME 2024 (Art of Problem Solving, 2024). Following prior work (Cui et al., 2025; AI@Meta, 2024), we employ greedy decoding for evaluation and report the accuracy metric. We compare P-GRPO against three baselines: (1) the original Qwen2.5-Math-7B model, (2) the base model trained via RL without thinking rewards, and (3) current SOTA mathematical models, including Llama-3.1-Instruct (AI@Meta, 2024), GPT-4o-2024-0806 (OpenAI, 2024), Eurus-2-PRIME (Cui et al., 2025), Qwen2.5-Math-Instruct (Yang et al., 2024).

P-GRPO effectively enhances the model's performance on mathematical tasks. As shown in Table 3, the base model with P-GRPO demonstrates comparable or superior performance compared to several

---

[2]https://gist.github.com/natolambert/1aed306000c13e0e8c5bc17c1a5dd300

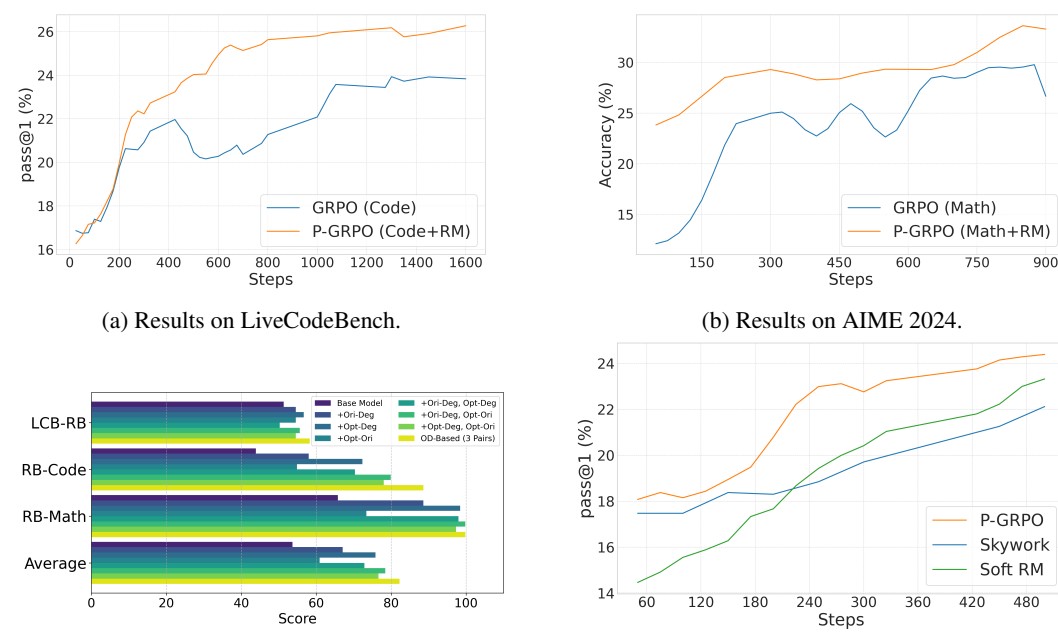

(a) Results on LiveCodeBench.

(b) Results on AIME 2024.

(c) The effectiveness of different preference pair sources for Qwen-2.5-Coder-7B.

(d) The performance of P-GRPO when integrated with different reward models.

Figure 4: Performance comparison of the model with P-GRPO against the GRPO baseline (a, b) and ablation studies on preference sources and reward models (c, d).

SOTA mathematical models. It demonstrates a 7.4% relative improvement over the RL baseline trained without thinking rewards. To further illustrate the superiority of P-GRPO, we analyze the performance trajectory on AIME24, as shown in Figure 4b. The results demonstrate that P-GRPO outperforms the baseline without thinking rewards throughout the training process, validating the generalization of our approach.

## 5 DISCUSSION

**Impact of Different Preference Pair Combinations** We train base models on different combinations with identical experimental settings. As illustrated in Figure 4c, models achieve optimal performance when trained on all types of preference pairs (due to space limitations, the results for 3B models are in Appendix A.5). This improvement can be attributed to the comprehensive learning signal provided by OD-based method, which enables the model to better distinguish reasoning processes. Notably, when training solely on Opt-Deg pairs, the model outperforms those trained on any other single pair type by 18.5% on average, demonstrating that a contrast in the quality of reasoning paths provides a clearer learning signal, enhancing the model's discriminative capabilities.

**Comparison with Other Reward Models** We replace the reward model in P-GRPO with Skywork-Reward-Llama-3.1-8B to further validate the effectiveness of our reward model. Due to computational constraints, we use Qwen2.5-Coder-7B-Instruct as the policy model and evaluate performance changes over the first 500 steps on LiveCodeBench. As shwon in Figure 4d, P-GRPO with our OD-based reward model demonstrates superior performance. This reveals that reward models trained solely on outcomes may inadvertently reinforce suboptimal reasoning patterns, as correct code does not necessarily reflect an optimal reasoning process (Chen et al., 2024). In contrast, our approach explicitly models the quality of intermediate reasoning steps, leading to more robust learning signals.

**The Impact of Reward Hacking** We introduce a soft reward formulation to demonstrate the susceptibility of RL to reward hacking. Specifically, we modify the reward as: $R_i = R_i^f + R_i^o + P_i^o \cdot R_i^t$, where $P_i^o$ denotes the pass rate of output $o_i$. We conduct experiments using Qwen2.5-Coder-7B-Instruct as the policy model and evaluate performance changes over the first 500 training steps on

LiveCodeBench. As shown in Figure 4d, the introduction of this soft reward computation consistently underperforms compared to P-GRPO, revealing that the reward model's rewards derived from erroneous code are inherently unreliable, and the policy model excessively exploits these noisy signals, leading to a performance degradation.

# 6 RELATED WORK

## 6.1 REINFORCEMENT LEARNING FOR LLM

RL enables LLMs to go beyond imitation learning and optimize generation based on task-specific rewards (Ouyang et al., 2022). This paradigm has been widely adopted across diverse natural language processing domains, such as code generation (Dong et al., 2024; Le et al., 2022; Liu et al., 2023). Proximal Policy Optimization (PPO) (Schulman et al., 2017) is a prevalent algorithm, valued for its stability. It optimizes the policy by using on-policy data, incorporating techniques such as Generalized Advantage Estimation (GAE) (Schulman et al., 2015) for variance reduction. More recent methods (Lin et al., 2025; Zhang et al., 2020) have sought to refine this process. For instance, GRPO (Shao et al., 2024) replaces the learned critic with an estimation of a baseline from group scores, calculating the relative advantage of each completion based on a rule-based reward function. Despite these advancements, they focus solely on evaluating the final generated code. Our work is predicated on the finding that the quality of this reasoning process has a profound impact on the final code's correctness. We introduce P-GRPO that explicitly rewards the reasoning process in conjunction with test case feedback to optimize the model.

## 6.2 REWARD MODEL EVALUATION ON REASONING TASKS

Evaluating the performance of reward models for reasoning tasks typically relies on verifiable problems. For example, the code subset of RewardBench (Lambert et al., 2024) utilizes HumanEval-Pack (Muennighoff et al., 2023), a multilingual extension of the HumanEval dataset. However, they typically focus only on the correctness of the final output, neglecting the quality of the intermediate reasoning process that produced it. To address these, we construct LCB-RB sourced from LiveCodeBench (Jain et al., 2025), which allows us to evaluate a reward model's discrimination capabilities on the intermediate steps of problem-solving.

# 7 CONCLUSION

In this paper, we introduce P-GRPO, a novel RL method designed to enhance the models' code generation capabilities. Our core innovation is to enrich the reward signals by combining rule-based rewards with rewards generated by a thinking reward model, which is trained with Optimized-Degraded based method. The thinking reward model achieves the best performance on LCB-RB, a benchmark constructed by us for discriminating between superior and flawed reasoning processes. Through extensive experiments across four code generation benchmarks, we demonstrated the effectiveness of our approach. Using a 7B model, P-GRPO achieves an average relative improvement of 4.5% on Pass@1 compared to the baseline with outcome-only rewards. This paradigm exhibits strong generalization capabilities when we apply it to the mathematical domain.

**Limitations and Future Work.** Our current experiments are constrained by computational resources. A natural progression is to apply P-GRPO to more powerful reasoning models such as DeepSeek-R1-Distill-Qwen-7B. This extension necessitates scaling our current output length from 4K to over 30K tokens and enhancing our data generation pipeline. Specifically, to build a reward model capable of evaluating such reasoning processes, the preference pairs must be generated by a powerful reasoning model. Beyond scaling to advanced reasoning models, another promising direction involves developing a self-sufficient, iterative learning framework. This would involve using the improved policy model itself to synthesize new preference pairs for training the next generation of the reward model.

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

## A  APPENDIX

### A.1  IMPLEMENTATION DETAILS

**Reward Model Setup**   We utilize Qwen2.5-Coder-32B-Instruct (Hui et al., 2024) to generate reasoning process preference data. The reward model is trained with a batch size of 128 and a learning rate of 1e-6 for 2 epochs. We partition the dataset using a 9:1 train-validation split and employ an early stopping strategy based on the validation set.

We compare our approach against several baselines: (1) Original Model: The base model without any additional fine-tuning. (2) SOTA Reward Models: Current best-performing reward models to validate the competitive performance of our method, including Starling-RM-34B (Zhu et al., 2023), EURUS-RM-7B (Yuan et al., 2024), Skywork-Reward-Llama-3.1-8B (Liu et al., 2024a), GPT-4-Turbo-2024-04-09 (Achiam et al., 2023), and GPT-3.5-Turbo-0125 (OpenAI, 2022). (3) Score-Based reward model: We employ Qwen2.5-Coder-32B-Instruct to score the initial reasoning processes based on the dimensions of reasoning quality, scoring prompt is in Appendix A.8, which is also employed by (Fan et al., 2025). We then perform SFT on the base model using consistent hyperparameter settings to those with the OD-based method.

### A.2  RL SETUP

For BigCodeBench, we use both the full and hard sets with complete configuration. For Live-CodeBench, we utilize the problems from October 2024 to February 2025, in line with prior work (Yang et al., 2025; Tian et al., 2025). We compare our approach against several baselines: (1) Original Model: the original model without any additional training. (2) SOTA Code Models: Current best-performing models on code generation tasks for competitive comparison, including Llama3-Instruct-70B (AI@Meta, 2024), Deepseek-Coder-V2-Lite-Instruct (Zhu et al., 2024), Qwen2.5-Coder-Instruct 14B (Hui et al., 2024), GPT-4-Turbo-2024-04-09 (Achiam et al., 2023), and GPT-3.5-Turbo-0125 (OpenAI, 2022) (3) SFT on RL Data: The model fine-tuned on the same dataset using identical hyperparameters with SFT. (4) RL without thinking reward: The model only with outcome and format rewards.

The RL training is conducted using VeRL (Sheng et al., 2024) on 8 NVIDIA A800 80GB GPUs, with a total batch size of 32 and a maximum output length of 4,096. We employ AdamW optimizer with a constant learning rate of 1e-6 and train for 1,600 steps. We remove the KL divergence term and adopt token-level policy gradient loss computation and the clip-higher mechanism with $\varepsilon_{\text{low}} = 0.2, \varepsilon_{\text{high}} = 0.28$ for training stability.

## A.3 PROMPT USED FOR RL TRAINING

As an AI Assistant, your task is to solve a user's question. First thinks about the reasoning process in the mind and then provides the user with the final answer. The reasoning process and answer are enclosed within <think> </think> and <answer> </answer> tags, respectively, i.e., <think> reasoning process here </think><answer> answer here </answer>.
**{problem}**
Write Python code to solve the problem. First, present your thinking process within <think> </think> tags. Then, present the code in a python code block within <answer> </answer> tags.

Figure 5: The Prompt used for RL training.

## A.4 SYNERGISTIC CORRELATION BETWEEN REASONING QUALITY AND CODE CORRECTNESS

To investigate the correlation between reasoning quality and code correctness, we employ a powerful LLM to generate multiple solutions with explicit reasoning traces for coding problems. We then utilize corresponding test cases to categorize the generated code into correct and incorrect implementations. To assess the quality of reasoning processes, we leverage GPT-4o-mini (OpenAI, 2024) to classify each solution's reasoning into three distinct categories: (1) flawless reasoning with consistent implementation, (2) flawed reasoning with consistent implementation, and (3) inconsistent reasoning and implementation. We exclude the third category from our analysis, as the misalignment between reasoning and implementation introduces confounding factors that would obscure the relationship between reasoning quality and code correctness. This filtering ensures that our study focuses specifically on cases where the implementation faithfully reflects the reasoning process, whether that reasoning is sound or flawed. Consequently, each generated output can be characterized by two attributes: (1) code correctness (correct or incorrect), and (2) reasoning quality (flawless or flawed).

To quantify the association between these two attributes, we perform the chi-square test (Pearson, 1900). Specifically, we utilise Qwen2.5-Coder-32B-Instruct with temperature $T = 1.0$ to generate 50 solutions for problems from LiveCodeBench v5. Our analysis yields a highly significant result with $p = 9.3 \times 10^{-15} \ll 0.001$, indicating a strong statistical dependence between reasoning quality and code correctness.

## A.5 IMPACT OF DIFFERENT PREFERENCE PAIR COMBINATIONS OF 3B MODELS

We present the performance of the 3B model under different pair combinations, as illustrated in Figure 6.

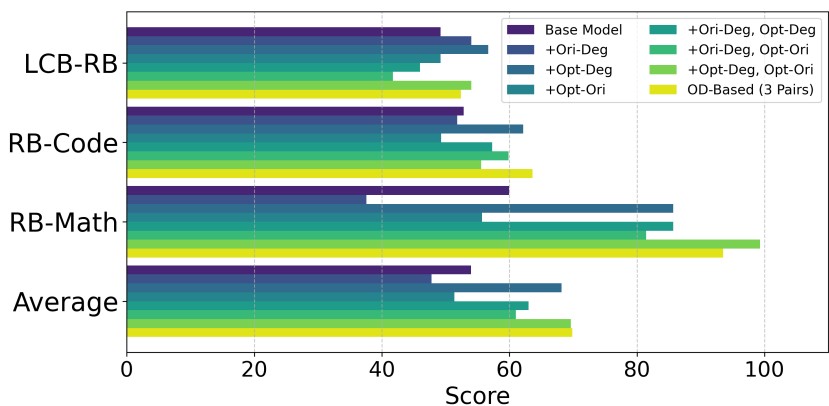

Figure 6: The effectiveness of different preference pair sources for Qwen-2.5-Coder-3B, where Ori, Deg, and Opt denote the original, degraded, and optimized reasoning paths, respectively.

864
865
866
867
868

## A.6 REASONING GENERATION PROMPT

869
870
871
872
873
874
875
876
877
878
879
880
881
882
883
884
885
886
887
888
889
890
891
892
893
894
895
896
897
898
899
900
901
902
903
904
905
906
907
908
909
910
911
912
913
914
915
916
917

---

**Initial Reasoning Generation Prompt**

# Task Objective
You are an Expert Problem Solver and Algorithmic Thinker. Your primary goal is to generate a detailed, step-by-step Chain-of-Thought (CoT) that deconstructs and logically solves the given problem. Your output should be the reasoning process itself, not the final solution or code.
# Input Data
[Problem Statement]
{problem_statement}
# Requirements for Your Reasoning
1. Deconstruct from First Principles: Begin by dissecting the problem statement. What is the core question? What are the explicit and implicit requirements? What are the inputs, outputs, and constraints? Break the problem down into smaller, more manageable sub-problems.
2. Analyze Examples and Edge Cases: Systematically use the provided examples and test cases to verify your understanding. Explicitly state what each test case teaches you.
3. Brainstorm and Strategize:
(1) Prioritize Optimal Approaches: Begin by brainstorming efficient strategies. First, explore algorithms and data structures that could lead to an optimal or near-optimal solution (e.g., hash maps, two-pointers, binary search, dynamic programming, greedy algorithms). Do not start by considering the brute-force approach.
(2) Select and Justify the Best Strategy: Evaluate the potential efficient approaches you've identified. Choose the most promising one and provide a clear justification for your choice. Analyze its trade-offs in terms of time complexity ($O(n)$), space complexity ($O(n)$), and implementation difficulty. For instance, "A hash map approach offers an optimal $O(n)$ time complexity at the cost of $O(n)$ space, which is an acceptable trade-off here. We will proceed with this strategy."
(3) Acknowledge Brute-Force as a Last Resort: Only if you determine that efficient algorithms are not applicable or are excessively complex to implement for the problem at hand, should you then articulate the reasoning for using a brute-force approach.
4. Develop a Step-by-Step Logical Plan: Based on your chosen strategy, create a clear, logical, and sequential plan.
(1) Mental Walkthrough: "Pre-run" your logic using a specific example. Narrate the state of your variables or data structures at each step of the plan.
(2) Refine and Self-Correct: After the walkthrough, reflect on the plan. Are there any logical gaps? Does it correctly handle all the identified edge cases? Could any step be simplified or made more robust? Acknowledge and address any flaws found during the mental walkthrough.
5. Clarity and Structure: Ensure the entire reasoning process is articulated in a clear, structured manner that is easy for a human to follow. The goal is to illuminate the *how* and *why* of the solution, not just the what.
# Output Format
Your response must contain ONLY the reasoning process, formatted in Markdown. Do not include any introductory or concluding remarks outside the reasoning block.

---

**Reasoning Degrading Prompt**

# Task Objective
You are a Red Teaming AI Agent specializing in crafting sophisticated negative training data for advanced reasoning models. Your task is to deliberately introduce a specific, targeted flaw into a 'Golden Chain-of-Thought' (CoT). This creates challenging examples that teach other models to identify and avoid logical errors.
# Input Data
[Problem Statement]
{question}
[Golden Chain-of-Thought]
{golden_CoT}
# Degradation Methods
1. Factually Incorrect Reasoning: Introduce a clear factual error into the logic. For example, misstate a core constraint from the problem, use an incorrect mathematical formula, or misrepresent the time/space complexity of a known algorithm.
2. Irrelevant or Misleading Path: Add steps that are factually correct on their own but are irrelevant to solving the actual problem. This creates a distracting and inefficient reasoning path.
3. Incomplete Reasoning: The reasoning starts correctly but halts before reaching the final step, leaving the logic unfinished and the conclusion unsupported.
4. Logical Gap / Jump: Remove a key intermediate step, making the jump from a premise to a conclusion seem abrupt and unsubstantiated, even if the final conclusion happens to be correct.
5. Chaotic or Acausal Reasoning: Invert the cause-and-effect relationship, or create a sequence of steps that are logically disconnected and do not follow a coherent progression.
# Execution Steps
1. Identify Methods: Identify one or more 'Degradation Methods' from the inputs (e.g., a comma-separated list like "Logical Gap, Factually Incorrect Reasoning").
2. Analyze & Plan: Carefully analyze the 'Golden CoT'. Strategically plan how to weave all the selected degradation methods into the reasoning. The flaws should be as subtle as realistically possible, modelling a plausible human error.
3. Generate Degraded CoT: Rewrite the CoT to create the flawed '[Degraded CoT]'. This section must contain ONLY the flawed reasoning itself.
4. Generate Explanation: Create a concise '[Explanation of Degradation]'. In this section, you must clearly list each degradation method you used, and for each one, pinpoint exactly how, where, and why you altered the original reasoning.
# Output Format
Your response MUST be in Markdown format and strictly adhere to the two-part structure below. If multiple degradations are applied, list each one in the explanation.
```markdown
[Degraded Cot]
(Write the Degraded Chain-of-Thought here.)
[Explanation]
(Describe where and how you applied the degradation method(s).)

**Reasoning Evolving Prompt**

# Task Objective
You are an AI Reasoning Optimizer, specializing in refining training data for advanced reasoning models. Your task is to take a Golden Chain-of-Thought (CoT) and apply one or more optimizations to make its logic more rigorous, efficient, and accurate. The goal is to create higher-quality training samples to elevate the performance of advanced reasoning models.

# Input Data
[Problem Statement]
{question}
[Golden Chain-of-Thought]
{golden_CoT}

# Optimization Methods
1. Factual Verification & Correction: Identifies and corrects a clear factual error within the reasoning. If no errors are found, this method should not be applied.
2. Focusing Logic: Identifies and removes any redundant steps from the original reasoning. This ensures every step directly contributes to the final goal, making the entire reasoning path more focused.
3. Comprehensive Reasoning: Extends a line of reasoning that may have halted prematurely or omitted final steps. This ensures the logical chain is fully closed and the conclusion is explicitly and robustly supported.
4. Bridging Logical Gaps: Adds necessary intermediate steps between logical nodes that seemed disjointed. This makes the transition from premise to conclusion smoother and more self-evident.
5. Enhancing Logical Flow: Reorganizes reasoning steps to follow a clearer, more intuitive causal or hierarchical order. This ensures the entire thought process is well-structured and flows seamlessly from start to finish.

# Execution Steps
1. Identify Methods: Based on the 'Optimization Methods' above, analyze the input Golden CoT and identify one or more specific methods for application (e.g., a comma-separated list like "Bridging Logical Gaps, Factual Verification").
2. Analyze & Plan: Carefully analyze the 'Golden CoT'. Formulate a clear strategy for integrating all selected optimization methods into the new reasoning process. The goal of the optimization is to make the reasoning more rigorous, clear, and persuasive.
3. Generate Optimized CoT: Rewrite the CoT to create the '[Optimized CoT]'. This section must contain ONLY the improved reasoning itself.
4. Generate Explanation: Create a concise '[Explanation of Optimization]'. In this section, you must clearly list each optimization method you used and, for each one, pinpoint exactly how, where, and why you improved the original reasoning.

# Output Format
Your response MUST be in Markdown format and strictly adhere to the two-part structure below. If multiple optimization methods are applied, list each one in the explanation.
```markdown
[Optimized CoT]
(Write the optimized Chain-of-Thought here.)
[Explanation]
(Describe where and how you applied the optimization method(s).)
```

## A.7 REASONING ASSESSMENT PROMPT

---

**Reasoning Flaw Assessment Prompt**

You are a top-tier code reviewer and logical analyst.

Your task is to rigorously analyze a programming solution by evaluating both its thought process ('<think>') and the consistency of its implementation ('<answer>').

Key Analysis Criteria:

1. Reasoning Soundness: Is the algorithm, logic, and step-by-step plan described in the '<think>' block a correct and robust way to solve the problem? Does this logic have flaws?

2. Implementation-Thought Consistency: Does the code in the '<answer>' block faithfully implement the logic described in the '<think>' block?

Input Format:

[Problem Description]

{problem_description}

[Solution]

{solution_content}

Your Task:

Strictly adhere to the following two-line output format.

Line 1: Output only 'Yes', 'No', or 'None' based on the following specific logic:

(1) Output 'Yes' ONLY if the reasoning in '<think>' has a flaw, AND the code in '<answer>' is a consistent implementation of that flawed reasoning.

(2) Output 'No' ONLY if the reasoning in '<think>' is sound, AND the code in '<answer>' is a consistent implementation of that sound reasoning.

(3) Output 'None' in all other scenarios. This primarily means any case where the code in '<answer>' is NOT a consistent implementation of the logic in '<think>', regardless of whether the reasoning is sound or flawed.

Line 2: Explain the reasoning for your judgment. Your explanation must address both the soundness of the thought process and its consistency with the final code.

---

## A.8 REASONING SCORING PROMPT

---

**Reasoning Scoring Prompt**

---

# Task Objective
You are an expert evaluator of AI reasoning. I will provide you with a problem and a candidate's chain-of-thought reasoning. Your goal is to judge the quality of this reasoning process and assign it a single score between 0 and 1. Your evaluation must focus on the logical integrity of the process, not merely on whether the final answer is correct.
# Input Data
[Problem Statement]
{question}
[Reasoning Process]
{reasoning_to_evaluate}
# Evaluation Criteria
1. Factual Errors: Does the reasoning introduce incorrect facts, misuse formulas, or misstate constraints from the problem?
2. Logical Gaps or Jumps: Are there missing steps? Does the conclusion jump from a premise without a clear, logical bridge?
3. Irrelevant or Misleading Paths: Does the reasoning include steps that, while perhaps factually correct, are irrelevant to solving the problem and create a distracting or inefficient path?
4. Incomplete Reasoning: Does the reasoning start correctly but stop short of reaching a final, supported conclusion?
5. Chaotic or Acausal Structure: Is the reasoning jumbled? Does it invert cause-and-effect or present steps in an illogical, disconnected order?
# Scoring Instructions
Provide a single score from 0, 0.1, 0.2,..., 1.0 based on the reasoning quality.
1.0: Perfectly sound reasoning. Clear, correct, complete, and efficient.
0.7 - 0.9: Minor flaws. Contains small, easily correctable errors or slight inefficiencies.
0.3 - 0.6: Significant flaws. Contains major logical gaps, factual errors, or irrelevant paths that seriously undermine the reasoning.
0.0 - 0.2: Completely flawed. The reasoning is chaotic, nonsensical, or fundamentally wrong from the start.
# Output Format
Be strict, you should only output the score without any explanation.

