# OpenReview forum: "Posterior-GRPO: Rewarding Reasoning Processes in Code Generation"
_ICLR.cc/2026/Conference — ICLR 2026 Conference Withdrawn Submission_

### Official Review · Reviewer_dof4 · 2025-10-30

**Soundness:** 2
**Presentation:** 2
**Contribution:** 2
**Rating:** 4
**Confidence:** 3

**Summary:**

This paper addresses the limitation that current reinforcement learning approaches for code generation focus on outcome signals (test pass rates) . The authors propose a three-part framework: (1) LCB-RB, a benchmark with 187 preference pairs for evaluating reward models on reasoning; (2) an Optimized-Degraded (OD-based) method for training reward models by systematically generating optimized and degraded versions of reasoning paths; and (3) Posterior-GRPO (P-GRPO), an RL algorithm that conditions process-based rewards on task success to mitigate reward hacking.

**Strengths:**

The paper identifies a genuine gap in current RL-based code generation approaches by highlighting that outcome-based rewards alone may miss important signals about reasoning quality.

The three-stage approach is well-structured and addresses complementary challenges—benchmark creation, reward model training, and RL algorithm design. The OD-based method for reward model training is more sophisticated than simple scoring approaches, using targeted degradation and optimization along specific reasoning dimensions (factual accuracy, logical rigor, logical coherence).

The paper demonstrates consistent improvements across multiple code benchmarks (HumanEval, MBPP, LiveCodeBench, BigCodeBench) and shows generalization to mathematical tasks (MATH500, Minerva Math, AIME24). The ablation studies examining different preference pair combinations and reward models add depth to the empirical analysis.

The paper is generally well-written with clear motivation, method description, and experimental setup. The visual examples (e.g., Figure 3) effectively illustrate the difference between reasoning with and without the thinking reward.

**Weaknesses:**

The core contribution of P-GRPO is essentially a gating mechanism that applies thinking rewards only when outcome rewards indicate success. While this is effective, it's a relatively straightforward modification to GRPO. The posterior conditioning strategy, though sensible, lacks theoretical justification for why this particular formulation is optimal. The paper would benefit from deeper analysis of alternative conditioning strategies and their trade-offs.

The paper cites DAPO but doesn't provide detailed comparison. DAPO also addresses the issue of uniform rewards by filtering rollout groups with identical rewards—conceptually related to P-GRPO's approach of using thinking rewards to differentiate among correct solutions. A thorough empirical and conceptual comparison with DAPO and other recent methods that enhance reward signals (e.g., filtering strategies, adaptive reward weighting) is needed. The related work section focuses heavily on describing prior work rather than positioning the contributions relative to these approaches.

While Section 5 includes an experiment showing that soft reward formulation (using pass rate P^o_i instead of binary R^o_i) underperforms, this is presented as an afterthought. The paper needs more rigorous analysis: What happens with intermediate conditioning strategies? How sensitive is performance to the threshold for applying thinking rewards?

**Questions:**

Can you provide a direct empirical comparison with DAPO under controlled conditions (same base model, same training data, same evaluation protocol)? How does P-GRPO's posterior conditioning compare conceptually and empirically to DAPO's filtering approach?

Can you report confidence intervals or perform significance tests for the main results? Are the improvements over the GRPO baseline statistically significant?

How does the reward model trained on DeepCoder-Preview-Dataset perform when applied to reasoning from models it hasn't seen during training (e.g., other model families)? Is there evidence of overfitting to the specific reasoning style of Qwen2.5-Coder?

---

### Official Review · Reviewer_14S7 · 2025-10-31

**Soundness:** 2
**Presentation:** 3
**Contribution:** 2
**Rating:** 2
**Confidence:** 3

**Summary:**

Posterior-GRPO introduces a refinement of the GRPO objective for verifiable reward with a reward model focused on the reasoning chain (CoTs) that is trained with a particular data augmentation methodology the authors call OD. The particular way in which the verifiable reward is combined with RM-based reward is that the RM-based reward is only applied if the final answer is correct, hence the "Posterior" in the "Posterior-GRPO". Authors demonstrated the effectiveness of RM trained with OD on a new LiveCodeBench derived RM benchmark (LCB-RB) and RewardBench, as well as the effectiveness of Qwen2.5-Coder trained with P-GRPO with the aforementioned RM.

**Strengths:**

Combining reward model with verifiable reward to increase reward granularity is an important topic that has received a lot of attention in recent years. However, the finer details of how they ought to be combined is not always obvious. The posterior conditioning of the RM reward based on final verifiable output is interesting and seems important.

While conceptually simple, the ablations seem to show that the method is quite effective at least for Qwen-2.5-coder-instruct, and, perhaps rather surprisingly, the approach is generalizable to MATH tasks despite the RM seemingly being trained only on code data.

Generally well written and easy to follow.

**Weaknesses:**

Novelty for the paper seems marginal. Data augmentation for RM training is standard practice so novelty on the specific detail for RM data curation is limited. Combining RM with verifiable reward, or other types of partial reward, is not new and has a long history of papers for code generation. Although the focus on reasoning chain is new given the recent advents of the corresponding reasoning models.

The ablations are not sufficient. From a modeling standpoint, since everything is done in the Qwen model family from policy used to generate RM data, RM base model, policy trained during RL, it is hard to gauge how well this method would generalize to other model families.

**Questions:**

In section 4.1, author argued for the effectiveness of P-GRPO in encouraging sound reasoning by comparing examples based on thinking reward score. Are these reward scores generated by the same thinking reward model that the model was trained on?

For clarification, how is the BT RM's output score bounded between 0-1? In my experience while the objective function may be bounded by the sigmoid in the BT objective, the decoded scalar reward is not necessarily bounded.

---

### Official Review · Reviewer_W5mD · 2025-11-01

**Soundness:** 2
**Presentation:** 3
**Contribution:** 2
**Rating:** 4
**Confidence:** 4

**Summary:**

This paper addresses a key limitation in reinforcement learning (RL) for code generation: most approaches rely only on outcome-based rewards (e.g., test pass rates) and ignore the reasoning process behind the generated code. The authors propose a three-part framework to incorporate process-level reasoning signals while avoiding reward hacking:
1. LCB-RB: a small benchmark contrasting “better” vs. “worse” reasoning traces for the same task.
2. OD-based reward modeling: an Optimized–Degraded data construction scheme that generates improved and degraded reasoning chains along factual accuracy, logical rigor, and coherence dimensions, training a Bradley–Terry reward model to discriminate reasoning quality
3. Posterior-GRPO: a GRPO variant that gates process rewards by correctness, only applying reasoning rewards when code passes all tests.

Experiments with Qwen2.5-Coder-7B-Instruct show consistent gains: average Pass@1 improves from 50.4 to 57.4, surpassing outcome-only GRPO and approaching GPT-4-Turbo. The OD-trained reward model matches GPT-4-Turbo on reasoning discrimination benchmarks and generalizes to math tasks.

**Strengths:**

1. OD‑based RM is practical: Training with contrastive optimized/degraded pairs is both conceptually sound and easy to reproduce; Table 2 shows clear gains over score-based models.
2. Consistent empirical improvements: P-GRPO improves or matches baselines across all four code benchmarks and generalizes to math.
3. Reproducibility: Training configs, prompts, and datasets are documented.

**Weaknesses:**

1. Limited evaluation scale: The main benchmark (LCB-RB) has only 187 pairs, which is far too small for robust generalization claims. Expanding it to at least several thousand pairs would strengthen the paper dramatically.
2. Reward-model artifacts: Because OD-pairs are produced by templated prompts using the same generator, the reward model might detect formatting differences rather than genuine reasoning quality. Cross-generator or human-annotated validation is missing.
3. Compute and efficiency overhead: P-GRPO requires both test-case verification and reward-model inference; no runtime or training-time cost comparison is given.
4. Potential contamination: Since LCB-RB derives from LiveCodeBench and the policy is also tested there, it would be important to ensure no data overlap or leakage.
5. No human verification of reasoning quality: The authors rely entirely on GPT-4o to judge reasoning soundness. A small human-rated subset would make the benchmark more credible.

**Questions:**

1. How many OD-triples were used in total for reward-model training beyond the 187 LCB-RB pairs?
2. If OD pairs are created by a different model family (e.g., DeepSeek-Coder) and different templates, how does RM accuracy and P‑GRPO performance change?
3. What fraction of sampled completions actually satisfy $R_o=1$ during RL training? Did posterior gating reduce early exploration?
4. Beyond the “soft reward” variant (Fig. 4d), can you show cases where the model exploited an ungated reward (e.g., verbose but wrong reasoning)?
5. Have you conducted a small-scale human audit to verify that GPT-4o’s “sound vs flawed reasoning” judgments are reliable? Inter-annotator agreement statistics would also be very helpful.

---

### Official Review · Reviewer_MzsY · 2025-11-01

**Soundness:** 2
**Presentation:** 2
**Contribution:** 2
**Rating:** 2
**Confidence:** 3

**Summary:**

The paper addresses key limitations in current methods for code generation: the decoupling of reasoning from the final outcome, the risk of reward hacking, and the absence of dedicated reasoning benchmarks and adaptive reward models. Initial empirical analysis confirms a significant statistical correlation between reasoning quality and code correctness.

To tackle these issues, the study introduces a unified framework centered on three innovations.

LCB-RB Benchmark: A new evaluation benchmark derived from LiveCodeBench v5, creating 187 "Superior/Inferior Reasoning" preference pairs to specifically assess the reasoning process.

OD-based Reward Model Training: A method that generates strong contrastive samples by optimizing and degrading reasoning paths based on factual accuracy, logical rigor, and coherence. The resulting 7B-parameter reward model achieves superior performance, outperforming models like GPT-4-Turbo in scoring reasoning quality.

Posterior-GRPO (P-GRPO) Algorithm: A novel anti-reward-hacking RL algorithm that conditions reasoning reward assignment on the code passing all test cases (posterior assignment). This strategy ensures reasoning optimization aligns with result correctness and overcomes the conventional GRPO's zero-gradient issue when all samples are correct.

Experimental results demonstrate strong performance in both code generation and mathematical reasoning. The paper concludes by detailing its contributions, noting resource limitations, and outlining future work like extending to long-text reasoning. All related models, data, and code are open-sourced.

**Strengths:**

- The core innovation of this work lies in the precise refinement and extension of the GRPO (Guided Reinforcement Learning with Policy Optimization) reward mechanism, notably through the introduction and assessment of reasoning process quality. Firstly, the authors establish a solid empirical foundation for reasoning evaluation, confirming a significant correlation between reasoning quality and final code correctness via a Chi-square test (with a $p$-value much less than $0.001$). Building upon this, the paper innovatively designs a reasoning process evaluation system based on three dimensions: factual accuracy, logical rigor, and coherence. This fundamentally minimizes cases where code is correct but the reasoning is flawed, shifting the model's optimization goal from merely achieving "correct results" to pursuing "logically sound reasoning."
- To effectively counteract the policy model generating seemingly plausible but useless reasoning to "game the reward," the paper constructs a hierarchical reward mechanism: the total reward is composed of a format reward, an outcome reward, and a reasoning reward. Crucially, the assignment of the reasoning reward is ingeniously designed as a posterior allocation—it is only granted if the code passes all test cases; otherwise, the reasoning reward is zeroed out. This design effectively suppresses the model's tendency to exploit useless reasoning for rewards. Furthermore, this mechanism addresses the inherent limitation of traditional GRPO, which suffers from a lack of learning gradient when all samples are correct, as the differentiation in reasoning quality continues to provide a meaningful gradient, guiding the model toward paths with superior reasoning.
- Addressing the scarcity of data for reasoning evaluation, the paper specifically develops the LCB-RB benchmark. This benchmark filters 880 problems from LiveCodeBench v5, generates preliminary "reasoning + code" samples using Qwen2.5-Coder-32B-Instruct, and employs GPT-4o for final manual screening and proofreading. The result is a high-quality dataset comprising 187 pairs of "superior vs. inferior reasoning." The authors also ensure the reliability of the evaluation results by using down-sampling techniques to balance the superior reasoning samples.

**Weaknesses:**

- The core design of P-GRPO, which stipulates that reasoning rewards are only granted upon correct outcomes, shares similar underlying principles with existing research where code correctness is tied to process-based rewards. However, the paper fails to clearly articulate the breakthrough innovation of this design in terms of algorithmic logic (e.g., compared to methods where outcome and process rewards are granted simultaneously) or reward calculation (e.g., the mathematical soundness or optimality of posterior reward allocation). This lack of clarity diminishes the perceived technical novelty of the approach.
- Crucial operational details of the OD-based method lack clear description. For instance, how is the quality of reasoning specifically judged across the three dimensions—"factual accuracy, logical rigor, and logical coherence"? How are redundant or logically disconnected steps precisely defined? Are there weight differences in the importance of these three dimensions? Furthermore, the specific prompts used to generate "better reasoning" or "worse reasoning," along with their design rationale, are not disclosed, posing significant difficulties for method replication.
- The representativeness and scope of applicability of the LCB-RB benchmark are limited, as the sample size is only 187 pairs and is solely derived from the single LiveCodeBench v5 dataset. Its generalization to other programming languages (non-Python) or different complexities of coding tasks (e.g., simple scripts vs. complex algorithms) remains unverified.
- The paper lacks an in-depth analysis of the potential issues of P-GRPO when handling complex reasoning scenarios. For example, when code passes all tests but the reasoning process contains "latent logical flaws".

**Questions:**

- Could the authors elaborate on the P-GRPO design of "assigning the reasoning reward only upon correct outcomes," specifically comparing it with existing reinforcement learning studies that also link code correctness and reasoning rewards? What is its unique technical breakthrough or performance gain in terms of core algorithmic logic, the mathematical advantage of posterior reward allocation, or anti-reward-hacking efficacy (e.g., the reduction in the model's reward-hacking probability compared to baseline methods on the same test set)? Can specific comparative analysis or experimental data be provided to support this?
- In the OD-based method, how are factual errors or redundant steps in the reasoning precisely identified and determined? Please provide specific prompt examples used to generate "better reasoning" or "worse reasoning" and explain the rationale behind their design. Furthermore, how are the three evaluation dimensions (factual accuracy, rigor, coherence) quantified or integrated into the prompt design to ensure the method's reproducibility?
- Given the small sample size and single source of the LCB-RB benchmark, do the authors plan to increase the sample size to over 500 pairs and incorporate data from diverse sources (e.g., different programming languages, varying code difficulty) to enhance the benchmark's representativeness? Could validation results on non-LiveCodeBench tasks (e.g., industrial code problems) be provided to demonstrate the accuracy and scope of applicability of the reward model on such tasks?
- Regarding the scenario where code passes all tests but the reasoning contains "latent logical flaws," how does P-GRPO's reward mechanism, particularly the calculation of the reasoning reward, identify these issues? Please clarify the granularity of the current reasoning quality evaluation (step-by-step or paragraph-level)? Can the mechanism effectively differentiate between "minor flaws" and "major defects"? Are there plans to introduce more detailed and specific evaluation dimensions in the future to enhance the detection capability for complex reasoning deficiencies?

---

### Note · Authors · 2025-12-03

I have read and agree with the venue's withdrawal policy on behalf of myself and my co-authors.